# Seed and Straw Characterization of Nine New Varieties of *Camelina sativa* (L.) Crantz

Iris Montero-Muñoz [1], David Mostaza-Colado [1], Aníbal Capuano [2] and Pedro V. Mauri Ablanque [1],*

[1] Departamento de Investigación Agroambiental, Instituto Madrileño de Investigación y Desarrollo Rural, Agrario y Alimentario (IMIDRA), Finca El Encín, Autovía A-2. Km. 38, 28805 Madrid, Spain

[2] Camelina Company S.L. Spain, Camino de la Carrera 11-11, Fuente el Saz de Jarama, 28140 Madrid, Spain

* Correspondence: pedro.mauri@madrid.org; Tel.: +34-91-887-94-03

**Abstract:** *Camelina sativa* (L.) Crantz is a promising oilseed crop that has increased worldwide attention because of its agronomic characteristics and potential uses. From an agricultural point of view, this plant can grow in different environments, providing a good yield with low input requirements. In addition, camelina seeds contain a high percentage of oil (36–47%) and protein (24–31%), making them interesting for food or energy industries. Nevertheless, its cultivation is not widespread in Europe, particularly in Spain. In the present context of global change and the search for new sustainable crops, we are conducting two pilot projects aiming to confirm that camelina is a good option for oilseed crops in semi-arid climates (especially in central Spain, Madrid) and to find new profitable varieties for farmers. To reach our objective we have used nine new varieties, recently developed, to characterize and compare their seed oil content, and their seed and straw chemical composition. Finally, with our preliminary results, we determine which varieties present better properties to be used in future agricultural research or breeding programs. These results are part of a larger study that we are carrying out.

**Keywords:** camelina; crops; oilseed; Spain; Brassicaceae



## 1. Introduction

Oilseed crops have great economic relevance worldwide, and occupy millions of hectares of land dedicated to agriculture. The high percentage of oils produced by their seeds and their agronomic characteristics allow them to be highly productive crops with low maintenance costs [1–3].

Within the groups of oilseeds, the Brassicaceae family stands out, in which we find numerous species that are used in the food industry (animal and human) as well as in the pharmaceutical and chemical industries. One of the oilseed species that has regained great interest in recent years is *Camelina sativa* (L.) Crantz due to its versatility [4]. This herbaceous annual with two biotypes (winter and spring; [5]), known as false flax, has been cultivated since the Late Stone Age and Middle Bronze Age in Scandinavia and Eastern Turkey [6]. The first hypotheses of the origin of its domestication suggested that it was in Europe or Western Asia, but recently, new studies propose the Caucasian region as the center of its domestication [7].

Morphologically [8], *Camelina sativa* is a highly branched plant, between 20−80 (−100) cm. It has a taproot and entire or dentate leaves; the basal ones form a rosette. The flowers are in terminal inflorescences, and they have yellow petals and four nectaries. The fruits are small silicles that contain numerous seeds (6–16) with a high oil content (36% to 47%; [9]).

The good agronomic characteristics of this oilseed can be of great interest in agriculture, which is currently becoming a serious problem for the environment and biodiversity [10]. *Camelina sativa* crops can help to mitigate this issue due to their great environmental adaptability; they resist drought and cold, and the attack of pests and/or pathogens, consequently reducing the number of inputs for their maintenance, especially irrigation,

fertilizers, and pesticides [6,11,12]. Its short life cycle (85–100 days), root system, and cold tolerance make it a good option for crop rotation in arid and semi-arid regions, minimizing fallow and proposing an alternative to monoculture [13,14]. According to different studies, crop rotation and intercropping produce ecosystem benefits, such as erosion reduction, soil organic matter improvement, and weed control [15–20]. The use of camelina in the process of rotation is a good alternative for crop diversification. This oilseed species compete with others, and although it has a lower yield than, for example, rapeseed, it has minor requirements and numerous potential uses than others [6,14,21,22].

Another benefit that camelina crops could have is related to biodiversity. The loss of flower-rich habitats due to intensive agriculture has led to a decline in the number of pollinators. *Camelina sativa* has nectariferous flowers and recent studies indicate the positive effects of growing this species on the diversity of insects, providing a healthy habitat for them (forage, nectar, nesting, etc.), and thus increasing the crop yield [3,23,24].

In addition to its ecosystem benefits and its agronomic characteristics, camelina stands out for its wide variety of uses. The high percentage of oils present in its seeds makes it a very suitable and interesting resource for the production of biofuels [25–29], and for the food industry [30]. Studies are also being carried out to explore its possible use in animal and human feed because of the protein richness of seed meal [31–35] and the great nutritional value of its seed oil [36,37].

Finally, in this context of climate and global change, camelina crops can play a very important role in reducing the use of fossil fuels, land use, and the generation of greenhouse gases, hence helping to make circular and sustainable agriculture that does not harm ecosystems or biodiversity [27,38–41].

Nowadays, despite its agronomic properties and its possible uses, camelina cultivation in Europe is not very widespread [23]. Therefore, within the framework of our projects "PDR18-CAMEVAR" and "FP22-CAMEPRO" [42–44], we aim to increase the knowledge about this promising crop and we intend to select a variety of *Camelina sativa* adapted to the central region of Spain that is profitable for farmers and sustainable in its production. Therefore, with this work, we expect to characterize the seed and straw of nine new varieties of camelina to determine their oil content, chemical composition, and production to explore their potential uses. These preliminary results can help in future breeding programs to find new varieties and innovative bio-based products.

## 2. Materials and Methods

The field experiments were carried out on three farms in the Community of Madrid (Spain), El Encín (Alcalá de Henares), La Chimenea (Aranjuez), and La Isla (Arganda), during two growing seasons 2019/20 and 2020/21 (Figure 1). As there are only slightly different environmental conditions, for this experiment we have not considered the effect of the environmental variability of the farms on each variety.

Camelina Company S.L. (Spain) has recently developed the nine commercial varieties used in this study. Five of them (V1, V2, V3, V4, and V13) were obtained by breeding and four (V6, V7, V8, and V11) were developed in grow chambers and selected by handpicking. The selection of varieties was based, mainly, on productivity and phenology, althoughchar-acteristicss such as seed weight, shattering or resistance to diseases have been taken into account. One of the varieties, V7, is a winter genotype while the other eight are spring genotypes.

To carry out the field experiment, we selected nine plots of one hectare of extension for each variety. Next, we added triple 15 fertilizer (NPK 15-15-15) using a dose of 200–250 kg/ha. We sowed between the second fortnight of November and the first week of December, using a conventional cereal seeder (seed dose between 8 kg/ha and 10 kg/ha).

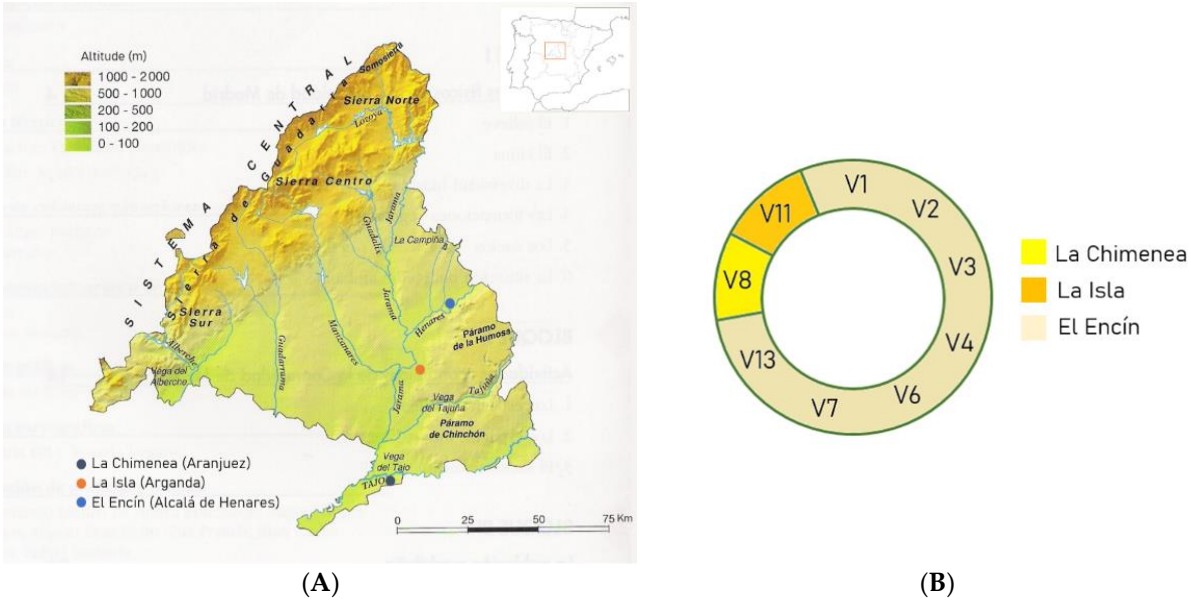

**(A)**　　　　　　　　　　　　　　　　　　　　　　　　　**(B)**

**Figure 1.** Location of the farms (**A**) and varieties used (**B**) in this study.

In January (one month after the sowing), during the rosette stage of the plant, we passed the roller again to guarantee compaction of the soil and avoid future problems with the harvester. At the beginning of winter (February) we fertilized the plots again with ammonium nitrosulfate (NSA) at a dose of 200–250 kg/ha. This fertilization happened just when the plant began to grow, to ensure the necessary nutrients for its growth and the proper development of the fruit.

The harvest was carried out at the beginning of July using a conventional harvester adjusted to the minimum air to reduce, as far as possible, the loss of seed. After harvesting each camelina variety, the machine was cleaned by harvesting barley, followed by a manual cleaning to avoid seed mixing. We stored the harvested material in 1000 kg bags in the facilities of Camelina Company S.L. There, they sieved and cleaned the samples using air and densimetric tables.

We performed chemical analyses only for the material harvested in the season 2020/2021, due to the coincidence of the 2019/2020 season with the COVID pandemic. Consequently, the characterization analyses of the seed and straw were outsourced and carried out at SERIDA (Regional Service for Agrifood Research and Development, Asturias, Spain); and the analysis of fatty acids at SETNA Nutrition and Services (Madrid, Spain). The fatty acid profile was obtained by gas chromatography and the chemical composition of the seed and the straw were obtained applying different techniques: calcination at 550 °C (ashes), Kjeldahl method (crude protein), extraction with petroleum ether (crude fat), Weende method (crude fiber), digestion in neutral buffer (neutral detergent fiber), digestion in acid buffer (acid detergent fiber), complexometric titration (calcium) and ultraviolet–visible spectroscopy (phosphorus).

Finally, we performed statistical analysis to compare the yield of the two seasons, using one-way ANOVA with appropriate post-hoc test and Student *t*-test using GraphPad Prism 9 (Graph Pad Software, San Diego, California, USA) and assuming a *p*-value below 0.05.

### 3. Results and Discussion

*3.1. Yield*

The varieties of camelina studied show an average yield of $1.29 \pm 0.36$ Tm/ha and $1.50 \pm 0.23$ Tm/ha during the 2019/20 and 2020/21 seasons respectively. In the first season, the yield ranged between 0.97 Tm/ha and 1.61 Tm/ha, and in the second season between 1.12 Tm/ha and 1.69 Tm/ha (Table 1). This means that in the second season, the production was higher than in the first one, although no statistically significant differences between

seasons (paired Student t-test) or varieties (one-way ANOVA and Tukey post hoc) were found. Thus, the variations between seasons may be due to differences in weather or soil conditions during crop growth.

**Table 1.** Yields of *Camelina sativa* varieties (season 2019/2020).

| Season | Farm | Variety | ha | Gross Weight (Tm) | Yield (Tm/ha) |
|---|---|---|---|---|---|
| 2019/20 | El Encín | V1 | 1.07 | 1.24 | 1.16 |
| | | V2 | 0.85 | 1.24 | 1.46 |
| | | V3 | 0.95 | 1.49 | 1.57 |
| | | V4 | 1.08 | 1.74 | 1.61 |
| | | V6 | 1.0 | 0.97 | 0.97 |
| | | V7 | - | - | - |
| | | V13 | 0.55 | 0.83 | 1.51 |
| | La Chimenea | V8 | 1.52 | 2.25 | 1.48 |
| | La Isla | V11 | 1.0 | 1.18 | 0.59 |
| | | | | Mean ± SD | 1.29 ± 0.36 |
| 2020/21 | El Encín | V1 | 0.66 | 1.11 | 1.69 |
| | | V2 | 1.45 | 1.62 | 1.12 |
| | | V3 | 0.40 | 0.62 | 1.55 |
| | | V4 | 0.80 | 1.35 | 1.69 |
| | | V6 | 0.45 | 0.75 | 1.67 |
| | | V7 | 0.65 | 0.89 | 1.36 |
| | | V13 | 1.45 | 1.66 | 1.14 |
| | La Chimenea | V8 | 1.35 | 2.16 | 1.60 |
| | La Isla | V11 | 0.95 | 1.58 | 1.67 |
| | | | | Mean ± SD | 1.50 ± 0.23 |

In the season 2019/20, the most productive varieties were V3 (1.57 Tm/ha) and V4 (1.61 Tm/ha), grown on the farm of El Encín (Alcalá de Henares). The variety less productive was V11 (0.59 Tm/ha), grown in La Isla (Arganda), but it should be noted that the low yield productivity of this variety was due to a rabbit plague that could not be treated due to the pandemic. In the season 2020/21 the most productive varieties were V1 (1.69 Tm/ha) and V4 (1.69 Tm/ha), and the less productive was V2 (1.12 Tm/ha) each one also grown in El Encín. The variety V7 was grown only in the second season and has a lower yield than other varieties (Table 1). Additionally, the results show that V4 is the variety that has the highest yield in both seasons.

If we compare the camelina production each season to other crops in Spain and the Community of Madrid (Figure 2A,B), the results indicate that camelina is more productive and profitable than other rainfed crops (wheat, barley, oats, etc.) and other oilseeds (rape or sunflower). Therefore, camelina could be a good option for farmers. Finally, the average yield of the camelina crop, 1.40 Tm/ha, is similar to that obtained in other studies carried out in different countries of the European Union [11,14,45–48], North America [49–52] and South America [53].

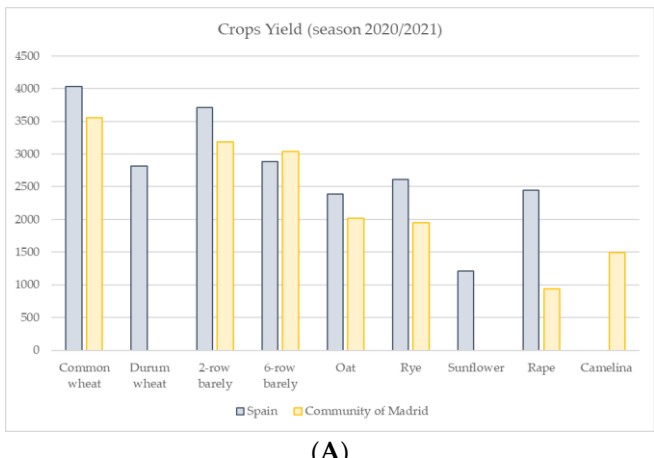 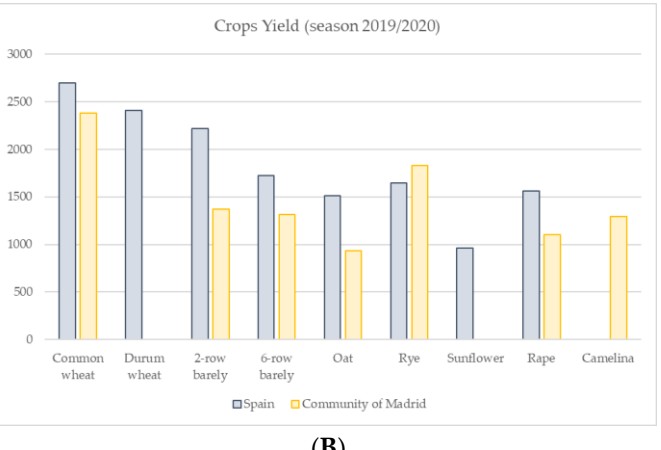

**(A)** **(B)**

**Figure 2.** Camelina yield compared to other crops in Spain and in the Community of Madrid (Data obtained from Ministry of Agriculture, Fisheries, and Food of Spain: Encuesta sobre Superficies y Rendimientos Cultivos).

*3.2. Camelina Oil Content and Fatty Acid Composition*

We found 23 fatty acids in the seeds of the different camelina varieties, although we did not find all of them in a single variety (Table 2).

**Table 2.** Fatty acid content (% based on dry seed mass) of camelina seeds varieties in the season 2020/21.

|  | V1 | V2 | V13 | V3 | V4 | V6 | V8 | V7 | V11 |
|---|---|---|---|---|---|---|---|---|---|
| C14:0 | 0.00 | 0.04 | 0.04 | 0.05 | 0.01 | 0.06 | 0.04 | 0.05 | 0.05 |
| C15:0 | - | 0.02 | 0.02 | - | 0.01 | 0.02 | 0.03 | - | 0.03 |
| C16:0 | 5.20 | 5.82 | 5.76 | 5.59 | 5.88 | 6.54 | 6.56 | 6.46 | 6.02 |
| C16:1 | 0.04 | 0.03 | 0.03 | 0.03 | 0.04 | 0.03 | 0.01 | 0.04 | 0.04 |
| C17:0 | - | 0.04 | 0.05 | - | 0.04 | - | 0.04 | - | 0.05 |
| C17:1 | 0.01 | 0.02 | - | 0.02 | 0.01 | 0.02 | 0.02 | 0.02 | 0.03 |
| C18:0 | 2.76 | 2.89 | 2.73 | 2.59 | 2.63 | 2.87 | 0.01 | 2.60 | 2.54 |
| C18:1 | 15.95 | 14.20 | 15.60 | 12.91 | 13.21 | 14.12 | 15.58 | 13.81 | 16.00 |
| C18:2 | 17.34 | 21.50 | 20.76 | 20.04 | 18.44 | 19.14 | 22.70 | 21.99 | 18.90 |
| C18:3 | 34.67 | 32.19 | 32.06 | 34.55 | 34.86 | 33.19 | 29.72 | 31.93 | 32.30 |
| Ratio C18:3/C18:2 | 2.00 | 1.50 | 1.54 | 1.72 | 1.89 | 1.73 | 1.31 | 1.45 | 1.71 |
| C20:1 | 15.35 | 14.25 | 13.94 | 14.89 | 14.38 | 14.84 | 13.92 | 13.52 | 14.91 |
| C20:2 | 1.57 | 2.00 | 1.90 | 2.00 | 1.97 | 1.87 | 1.69 | 1.77 | 1.58 |
| C20:3 | - | 1.20 | 1.22 | - | - | - | 1.01 | - | 1.17 |
| C20:4 | 0.26 | 0.28 | 0.27 | 0.26 | 0.24 | 0.29 | 0.34 | 0.34 | 0.32 |
| C22:0 | 1.29 | - | - | 1.37 | 1.49 | 1.39 | - | 1.15 | - |
| C21:0 | 0.04 | 0.02 | 0.02 | 0.03 | 0.02 | 0.02 | 0.01 | 0.02 | - |
| C22:1 | 2.85 | 2.68 | 2.79 | 3.16 | 3.17 | 3.02 | 2.77 | 3.17 | 3.32 |
| C22:2 | 0.12 | 0.15 | 0.14 | 0.17 | 0.16 | 0.15 | 0.15 | 0.16 | 0.16 |
| C22:5 | - | - | - | - | 0.03 | 0.01 | - | - | - |
| C22:6 | 0.04 | - | 0.01 | 0.01 | - | 0.04 | 0.02 | 0.02 | 0.02 |
| C24:1 | 0.77 | 0.69 | 0.70 | 0.54 | 0.69 | 0.61 | 0.67 | 0.62 | 0.67 |
| C23:0 | 0.04 | 0.05 | 0.04 | 0.05 | 0.05 | - | 0.05 | 0.06 | 0.04 |
| C24:0 | 0.34 | 0.15 | 0.21 | 0.43 | 0.28 | 0.43 | 0.19 | 0.42 | 0.26 |
| Saturated | 9.69 | 9.04 | 8.86 | 10.10 | 10.39 | 11.32 | 9.97 | 10.76 | 8.99 |
| Monounsaturated | 34.99 | 31.87 | 33.07 | 31.55 | 31.49 | 32.65 | 32.96 | 31.18 | 34.97 |
| Polyunsaturated | 53.98 | 57.32 | 56.36 | 57.04 | 55.69 | 54.67 | 55.62 | 56.21 | 54.46 |
| Ratio Mono/Poly | 0.65 | 0.56 | 0.59 | 0.55 | 0.57 | 0.60 | 0.59 | 0.56 | 0.64 |

Nine of them were saturated fatty acids (myristic acid (C14:0), pentadecanoic acid (C15:0), palmitic acid (C16:0), heptadecanoic acid (C17:0), stearic acid (C18:0), heneicosa-

noic acid (C21:0), behenic acid (C22:0), tricosanoic acid (C23:0) and lignoceric acid (C24:0)) and 12 unsaturated fatty acids. Among them, six were monounsaturated (palmitoleic acid (C16:1), cis-10-heptadecenoic acid (C17:1), oleic acid (C18:1), eicosenoic acid (C20:1), erucic acid (C22: 1), nervonic acid (C24: 1)), and eight were polyunsaturated (linoleic acid (C18: 2, n-6), α-linolenic acid (C18: 3, n-3), cis-11,14-eicosadienoic acid (C20:2), eicosatrienoic acid (C20: 3), cis-13,16 docosadienoic acid (C22:2), arachidonic acid (C20:4), docosapentaenoic acid (DPA) (C22:5), and docosahexaenoic acid (DHA, C22:6)). The top five fatty acids found in all camelina varieties were α-linolenic, linoleic, oleic, eicosenoic and palmitic acid. In most varieties, polyunsaturated fatty acids (PUFA) were present up to 53–57%, monounsaturated up to 31–34%, and saturated up to 8–11%. The slight differences in fatty acid composition between varieties may be due to their genetic background, soil quality, and/or weather conditions [54].

Regarding the varieties, there are no great changes among them in the content of fatty acids (Table 2). V8 and V11 were the richest, containing 22 of the 23 fatty acids found in camelina seeds and presenting a high percentage of polyunsaturated and monounsaturated fatty acids and, accordingly, a lower percentage of saturated fatty acids. These two varieties were grown in La Chimenea and La Isla respectively.

All the varieties are rich in PUFA (n-6/n-3), with V2 presenting the maximum percentage. The highest content of linolenic acid (34.86%) and erucic acid (3.17%) was found in V4, and V2 has the lowest percentage of erucic acid (Table 2).

Due to the high content of fatty acids, camelina oil is a promising option for biodiesel production or food use [25–29,55]. Nevertheless, the varieties of camelina with a high content of PUFA, especially linoleic and linolenic, are not appropriate as biofuels because the oxidative stability of the biodiesel obtained is very poor [28]. Thereby, V11, the variety with the lowest content of linolenic and linoleic acids (51.2 %), is potentially suitable to be considered for biofuel production. Furthermore, camelina V11 (along with V1) presents the highest ratio of monounsaturated to polyunsaturated fatty acids (Figure 3), a must-have condition to be competitive as a biofuel [56,57]. In this sense, obtaining camelina biodiesel has lower requirements and produces less greenhouse gas emissions than other crops [27], making it an environmentally interesting option. In all, camelina could be considered a suitable option for biofuel production.

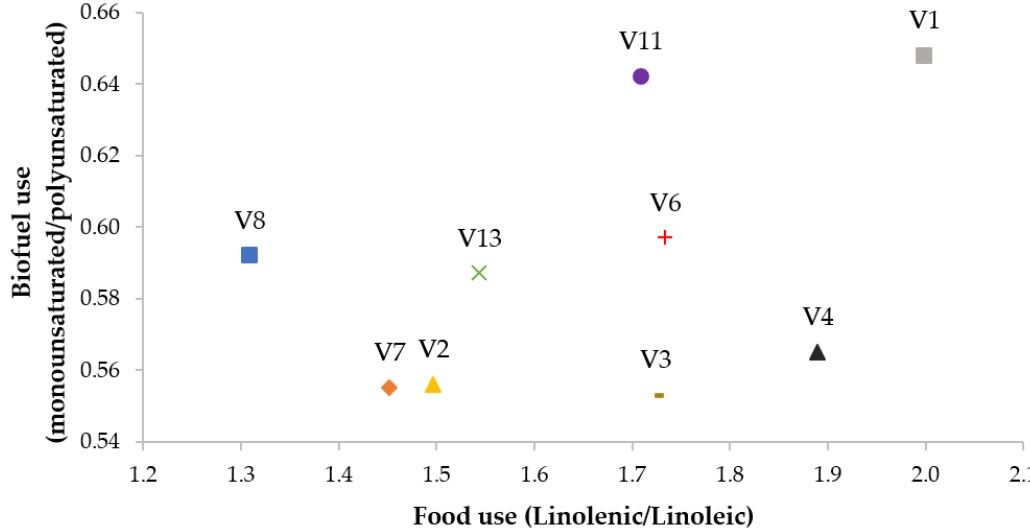

**Figure 3.** Presentation of nine varieties of *Camelina sativa* reflecting their potential use for food (linolenic/linoleic acid) and biofuel (monounsaturated fatty acids/ polyunsaturated fatty acids).

Our results indicate that some varieties of camelina can be used for food purposes (Figure 3). Regarding this potential food use, camelina oil should have a balanced linolenic/ linoleic acid ratio and a saturated fatty acid content below 10% [57,58]. Also, in Europe,

the content of erucic acid must be <5% [59]. Consequently, as variety V13 has a good ratio linolenic/linoleic, a low percentage of erucic acid (2.79%), and the lowest content in saturated fatty acids (8.86%), the oil of this variety could be a candidate for human consumption.

The results obtained in this work confirm that the fatty acid content in *Camelina sativa* seed oil has a high amount of unsaturated (89%), polyunsaturated and monounsaturated fatty acids, especially linolenic (C18:3; 32%), linoleic (C18:2; 20%), oleic (C18:1; 15%) and ecosienoic (C20:1; 14.01%), and a lower percentage of saturated fatty acids (9.9 %), in agreement with other studies [6,28,46,54,55,60,61]. In addition, camelina seed oil composition includes other minor fatty acids such as palmitic (C16:0; 6%), stearic (C18:0; 2.4%), and erucic (C22:1; 3%) acids. The fatty acid profile of each variety might differ because of their different cultivation conditions, genotype, location, environmental conditions, and fertilizer inputs [62,63].

Therefore, from a nutritional perspective, the fatty acid composition of camelina oil is important because it has a high content of linolenic and linoleic acid, both included as omega 3-6 fatty acids, which help in human or animal diet, increasing its nutritional value. Ratusz et al. [37] confirmed that camelina oil, due to its high nutritional value, could be a new resource for the pharmaceutical and human food industry.

Comparing the oil content and fatty acids composition of Camelina sativa seeds with other oil crops, we can infer that camelina is a better option for food use. In comparison with other crops, such as palm, kernel, soybean, sunflower, or canola (Table 3), it has the highest content of linolenic acid and a higher percentage of linoleic than canola. Regarding saturated fatty acids (Table 3), camelina oilseed has the lowest content in this kind of acids together with the sunflower. The oilseed of LEAR canola oil has a fatty acids profile similar to C. sativa, but the erucic acid content, considered a limiting factor for human consumption, in the oil of C. sativa ranged from 2.68% to 3.17%, while in the LEAR canola is less than 2% [64]. Nevertheless, this makes camelina a potential option for human consumption.

**Table 3.** Typical fatty acid composition in major oil crops and camelina (%).

|  | [a] Palm | [a] Kernel | [a] Soybean | [a] Sunflower | [b] Canola | [c] Camelina |
|---|---|---|---|---|---|---|
| C16:0 | 44 | 8 | 10 | 6 | 4.5 | 6.0 ± 0.5 |
| C18:0 | 4 | 2 | 4 | 5 | 2.4 | 2.4 ± 0.9 |
| C18:1 | 39 | 16 | 18 | 22 | 61.8 | 14.6 ± 1.2 |
| C18:2 | 10 | 3 | 55 | 66 | 17.3 | 20.1 ± 1.8 |
| C18:3 | <0.5 | <0.5 | 13 | <0.5 | 6.8 | 32.8 ± 1.7 |

[a] [65]; [b] [66]; [c] Data from this work.

### 3.3. Straw and Seed Chemical Composition

Regarding the seed and straw of *Camelina sativa*, we present the chemical composition: ashes, dry matter (DM), crude fat, crude protein (CP), crude fiber, metabolizable energy, digestibility, net energy for lactation, neutral detergent fiber (NDF), acid detergent fiber (ADF), calcium and phosphorus) of our cultivated varieties in Table 4 (straw) and Table 5 (seed).

**Table 4.** Nutrient composition of *Camelina sativa* straw.

|  | V1 | V2 | V13 | V3 | V4 | V6 | V8 | V7 | V11 |
|---|---|---|---|---|---|---|---|---|---|
| Dry matter (%) | 89.69 | 90.18 | 90.38 | 89.91 | 90.13 | 90.53 | 90.09 | 90.48 | 90.76 |
| Ashes (% dsm) | 4.83 | 3.83 | 3.58 | 6.06 | 5.52 | 4.62 | 4.65 | 4.99 | 3.89 |
| Crude protein (% dsm) | 8.08 | 7.72 | 6.38 | 8.27 | 9.44 | 9.43 | 5.49 | 6.80 | 5.64 |
| Crude fat/ether (% dsm) | 0.88 | 0.65 | 0.67 | 0.68 | 1.12 | 0.98 | 0.62 | 0.72 | 0.47 |
| Crude fiber (% dsm) | 51.30 | 54.01 | 55.89 | 51.92 | 49.92 | 51.10 | 56.97 | 53.29 | 57.91 |
| Neutral detergent fiber (% dsm) | 78.33 | 80.45 | 83.65 | 76.62 | 76.19 | 76.33 | 81.03 | 79.05 | 82.82 |
| Acid detergent fiber (% dsm) | 59.57 | 61.00 | 63.73 | 59.58 | 58.11 | 60.04 | 63.13 | 60.67 | 63.85 |
| Digestibility (%) | 36.21 | 36.19 | 34.51 | 38.55 | 37.76 | 37.75 | 35.86 | 31.11 | 33.15 |
| Metabolizable Energy (MJ/kg MS) | 4.82 | 4.87 | 4.66 | 5.07 | 4.99 | 5.04 | 4.79 | 4.80 | 4.46 |
| Net Energy for lactation (NRC89; Mcal/kg MS) | 0.69 | 0.69 | 0.66 | 0.73 | 0.71 | 0.72 | 0.68 | 0.68 | 0.63 |

**Table 5.** Nutrient composition (%) of *Camelina sativa* seed.

|  | V1 | V2 | V13 | V3 | V4 | V6 | V8 | V7 | V11 |
|---|---|---|---|---|---|---|---|---|---|
| Dry matter (%) | 91.54 | 92.25 | 92.34 | 93.24 | 93.94 | 93.76 | 94.68 | 94.07 | 92.78 |
| Ashes (% dsm) | 4.20 | 3.73 | 3.69 | 3.76 | 3.82 | 4.19 | 4.10 | 4.34 | 4.93 |
| Crude protein (% dsm) | 28.38 | 27.77 | 27.07 | 28.08 | 27.87 | 27.17 | 27.66 | 29.96 | 28.78 |
| Crude fat/ether(% dsm) | 39.17 | 40.47 | 39.12 | 40.34 | 41.22 | 41.03 | 37.90 | 36.72 | 36.18 |
| Crude fiber (% dsm) | 14.82 | 13.16 | 17.46 | 15.78 | 15.03 | 12.02 | 18.92 | 15.15 | 13.70 |
| Calcium (% dsm) | 0.31 | 0.30 | 0.26 | 0.29 | 0.28 | 0.29 | 0.27 | 0.26 | 0.51 |
| Phosphorus (% dsm) | 0.69 | 0.60 | 0.60 | 0.64 | 0.66 | 0.68 | 0.70 | 0.77 | 0.75 |

*Camelina sativa* straw is nutritionally rich having DM content between 89 and 90%. The CP content is 7.47%, while camelina seed DM content ranges from 91 to 94%, and the CP from 27 to 29%. The crude fat concentration ranges between 0.47 and 1.12% in camelina straw and between 36 and 41% in camelina seeds. The net energy for lactation (NEL) of camelina straw ranges between 0.66 and 0.73 Mcal/kg, the digestibility between 31.11 and 38.55 %, and the metabolizable energy (ME) between 4.46 and 5.07 MJ/kg. Crude fiber (CF) content of camelina straw ranges from 49.92 to 57.91%, while camelina seed CF content is between 12.02 and 18.92%. The ash content of camelina straw has been reported between 3.58 and 6.06%, and in camelina seeds between 3.69 and 4.93%. The acid detergent fiber (ADF) ranges between 14.4% and 25.4%, whereas neutral detergent fiber (NDF) ranges between 19.8 and 49.5% in camelina straw. Moreover, in camelina seeds, we have found some minerals such as calcium (0.26-0.51%), and phosphorus (0.60–0.77%).

Crude fiber (CF) content of camelina straw ranges from 49.92 to 57.91%, while camelina seed CF content is between 12.02 and 18.92%. The ash content of camelina straw has been reported between 3.58 and 6.06%, and in camelina seeds between 3.69 and 4.93%. The acid detergent fiber (ADF) ranges between 14.4% and 25.4%, whereas neutral detergent fiber (NDF) ranges between 19.8 and 49.5% in camelina straw. Moreover, in camelina seeds, we have found some minerals such as calcium (0.26–0.51%), and phosphorus (0.60–0.77%).

The use of camelina seeds and straw is an opportunity to promote environmental sustainability and lower costs. Straw or seed could be an excellent addition to diets for ruminants [35]. In this sense, regarding the straw chemical composition of each variety (Table 4), we can see that V3, V4, and V6 present higher values of digestibility (38.55, 37.76, 37.75% respectively) and CP (8.27, 9.44, 9.43% respectively), and the lowest values of neutral NDF. Moreover, the variety which has the lowest content of ashes is V13, having a potential energetic valorization [67]. Camelina seed is nutritionally rich too. It has a high DM content, a high level of crude fat (oil and fatty acids), and minerals (Table 5). V8 is the variety that has the highest content in DM (94.68%), and varieties V4 and V6 present high values of crude fat (41.22 and 41.03% respectively) and low values of crude fiber (15.03

and 12.02% respectively). This means that these varieties could be suitable for animal feed. Finally, the variety with the lowest ash content is V13 (3.69%).

Comparing other oilseeds with camelina, camelina has a lower crude protein (CP) content (28.08%) than soybean meal (49.6%) and sunflower meal (39.3%) but is similar to canola meal (34.5%) [35,68]. Camelina meal contains about 15% crude fiber (CF) mostly accounted for by cellulose. The most EE content was in expeller-extracted camelina meal (CE) (13.5%) [60]. Moreover, we are analyzing camelina hulls but we do not have conclusive results yet. Previous studies [35,69] have seen that camelina hulls have a similar NDF and ADF content, as canola, and a high ADL, and lower CP than soybean [35,60]. Finally, some studies have shown that camelina has an average of 45–47% crude protein in the meal and 10–11% of fiber, similar to soybean [70]. This means that higher percentages of crude protein improve the quality of the meal. Some researchers observed that camelina oil could improve the fatty acid content in milk and meat [71,72]. Overall, camelina could be an alternative to traditional oilseed crops such as canola, soybean, and maize in animal diets [73,74].

Our results show that camelina is a promising oilseed crop with many potential uses: biofuel, animal feed, the human feed industry, and feedstock. In Spain, mainly in the center, agronomic research is needed to improve the production system and help to develop camelina as a commercial oilseed crop.

## 4. Conclusions

In this context of global change, agricultural production can play a very important role in reducing the use of fossil fuels, in land use, and in the generation of greenhouse gases, hence helping to make agriculture circular and sustainable. Finding new crops or varieties more adapted to the current climate could be a challenge for agriculture but could help to make it more sustainable and profitable for farmers. Camelina sativa possesses good agronomic properties and many studies show the potential uses of its oil. This work allowed identifying some camelina varieties, such as V1 and V4, which could be considered good candidates for improving seed yield, but also others, such as V11, highly suitable for improving seed oil content, V3 for animal food, and V13 for energetic valorization for biomass production. In addition, we confirm that camelina can become a profitable alternative to canola in Spain. These preliminary results may assist future experiments to develop higher-yielding varieties for several uses (increase seed oil content, crude protein, minerals, ashes, etc.) and adapt to different environmental conditions.

**Author Contributions:** Conceptualization, D.M.-C. and P.V.M.A.; methodology, D.M.-C. and A.C.; validation, P.V.M.A. and D.M.-C.; formal analysis, D.M.-C.; investigation, P.V.M.A.; resources, A.C.; data curation, D.M.-C.; writing—original draft preparation, I.M.-M.; writing—review and editing, I.M.-M. and D.M.-C.; visualization, P.V.M.A.; supervision, D.M.-C.; project administration, D.M.-C. and P.V.M.A.; funding acquisition, P.V.M.A. and D.M.-C. All authors have read and agreed to the published version of the manuscript.

**Funding:** This research was funded by European Union through the European Agricultural Fund for Rural Development (EAFRD), the Ministry of Agriculture, Fisheries and Food of Spain, and the Community of Madrid through IMIDRA within the framework of the PDR-CM214-202, project number PDR18-CAMEVAR. FP22-CAMEPRO is funded by the Community of Madrid through IMIDRA.

**Data Availability Statement:** The data presented in this study are available on request from the corresponding author. The data are not publicly available due to restrictions e.g., privacy.

**Conflicts of Interest:** The authors declare no conflict of interest.

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
