# Peer review of "Seed and Straw Characterization of Nine New Varieties of Camelina sativa (L.) Crantz"

_land, doi:10.3390/land12020328_

Round 1

Reviewer 1 Report (Previous Reviewer 1)

Thank you for considering my comments in the previous review.

In my opinion, Your revised manuscript is now "richer, fuller" and more interesting for readers.

Author Response

REVIEWER #1 Thank you for considering my comments in the previous review. In my opinion, Your revised manuscript is now "richer, fuller" and more interesting for readers. Thanks to reviewer 1 for revising the manuscript a second time.

Reviewer 2 Report (New Reviewer)

The authors have evaluated the yield and several seed quality traits of oilseed crop Camelina Sativa in Spain. By presenting the results in the context of agricultural productivity and sustainability, this manuscript does have merits for camelina breeding and development. However, multiple issues need to be addressed for better communication and scientific publication.

1.    The citations of this manuscript need to be carefully checked. Many statements were made without citing the primary studies. For example, ref. [4] is not the primary study for camelina oil quantification; Line 43 obviously lacks a citation for camelina flower morphology; Camelina spring types and several winter types have been extensively evaluated in a world-wide collection of germplasms/accessions (doi.org/10.1002/tpg2.20110;  doi.org/10.1534/g3.119.400957) at phenotypic, genotypic, and genomic levels, which should be compared or discussed in multiple related places in the context. In short, as a brief communication, this manuscript has room for improving its references on the most updated and relevant research of camelina.

2.    Many points in the text are vaguely presented. For instance, “good agronomic characteristics” in line 46; “biodiversity” in line 60; “ecosystem benefits” in line 65; “circular and sustainable agriculture” in line 74, etc. All those points should be made more specifically and clearly.

Below are my comments and suggestions on specific lines,

Line 44, ‘siliques’ may be the correct word.

Line 47. This sentence needs a rewrite. ‘agriculture’ itself is not a serious problem.

Line 48-51 grammar error of the whole sentence.

Line 60-64 the whole paragraph is about biodiversity conservation, not the biodiversity of camelina. Please be specific.

Line 66 delete ‘that can be given to it’.

Line 68 write concisely. For example, change to “……because of the richness of its seed flour protein and the great nutritive value of its seed oil.”

Line 93 change “discovered” to “developed”.

Line 113 move the first comma to the end of ‘variety’.

Line 117-127 the detail and related citations of analytic methods are missing.

Line 148 change “the low productivity found in the yield” to “the low yield productivity”.

Line 158-159 delete ‘profitable’. How did we know camelina is more profitable with only yield data compared here?

Line 210 Please briefly present why the content of erucic acid must be <5% in Europea.

Line 308 global change?

Author Response

REVIEWER #2 The authors have evaluated the yield and several seed quality traits of the oilseed crop Camelina Sativa in Spain. By presenting the results in the context of agricultural productivity and sustainability, this manuscript does have merits for camelina breeding and development. However, multiple issues need to be addressed for better communication and scientific publication. Thanks to reviewer 2 for revising the manuscript. We have answered (in red) to all the comments made by the reviewer. 1. The citations of this manuscript need to be carefully checked. Many statements were made without citing the primary studies. For example, ref. [4] is not the primary study for camelina oil quantification; Line 43 obviously lacks a citation for camelina flower morphology; Camelina spring types and several winter types have been extensively evaluated in a worldwide collection of germplasms/accessions (doi.org/10.1002/tpg2.20110; doi.org/10.1534/g3.119.400957) at phenotypic, genotypic, and genomic levels, which should be compared or discussed in multiple related places in the context. In short, as a brief communication, this manuscript has room for improving its references on the most updated and relevant research of camelina. Camelina description is an original one, made by mixing different descriptions, but following the suggestions made by reviewer 2, we have added a citation. We have not compared genomic data since it is work that we are working on. The two provided articles are about the camelina genome, but this communication is about the chemical composition of seeds and the straw and oil content of seeds. Now, we are planning to genotype these varieties to develop the most interesting ones but it is out of the scope of this communication. We consider that the citations included are relevant and introductory to the submitted manuscript. For example, ref [4] is from 2020 in order to manifest that it is a subject of scientific interest nowadays. 2. Many points in the text are vaguely presented. For instance, “good agronomic characteristics” in line 46; “biodiversity” in line 60; “ecosystem benefits” in line 65; “circular and sustainable agriculture” in line 74, etc. All those points should be made more specifically and clearly. The points cited by the reviewer are, in our consideration, terms constantly used in scientific literature and even familiar to the general public. They have been used in order to contextualize our work, so we believe that it is not necessary to explain them in further detail. Below are my comments and suggestions on specific lines, Line 44, ‘siliques’ may be the correct word. The word silicle is correctly used: a silicle is a seed vessel resembling a silique, but about as broad as it is long. This definition or a similar one can be found in any botanical dictionary. Line 47. This sentence needs a rewrite. ‘agriculture’ itself is not a serious problem. We disagree. Agriculture is a problem in the way that it is currently developed: e.g. the use of chemicals, large monocultures, aquifer contamination, etc. The reviewer can find many publications about problems caused by agricultural systems e.g., Tsoraeva et al. 2020, Lanjewar 2022, Zhou et al. 2022, Nat. Geo. 2022/https://education.nationalgeographic.org/resource/environmental-impacts-agricultural-modifications Line 48-51 grammar error of the whole sentence. The sentence has been rephrased to improve clarity. Line 60-64 the whole paragraph is about biodiversity conservation, not the biodiversity of camelina. Please be specific. We think that the paragraph is clear, it refers to biodiversity. We talk about the benefits of camelina crops to biodiversity. Biodiversity does not refer to varieties of crops. Line 66 delete ‘that can be given to it’. Corrected. Line 68 write concisely. For example, change to “……because of the richness of its seed flour protein and the great nutritive value of its seed oil.” Corrected: Studies are also being carried out to explore its possible use in animal and human feed because of the protein richness of its seed meal [31-33] and the great nutritional value of its seed oil [36, 37]. Line 93 change “discovered” to “developed”. Changed. Line 113 move the first comma to the end of ‘variety’. Done. Line 117-127 the detail and related citations of analytic methods are missing. The methods and experimentation were carried out in approved laboratories outside our center. We have cited the techniques used for the extraction of each chemical component. Line 148 change “the low productivity found in the yield” to “the low yield productivity”. Done. Line 158-159 delete ‘profitable’. How did we know camelina is more profitable with only yield data compared here? We think it is profitable because the yield is good and the maintenance of the crop is cheap. So it is profitable for farmers, but we have changed profitable for a good option. Line 210 Please briefly present why the content of erucic acid must be

Reviewer 3 Report (New Reviewer)

Dear author(s),

there are some inspiring insights thorough the manuscript and I tend to agree on its publication. However, there are few points that needs to be quickly addressed to improve its overall communication:

Title:

1/ summarise in a clear statement why your work is important to humanity

Abstract:

2/ strictly follow the established schema of writing academic Abstract: A/ introduction (urgency and significance of the research hypothesis); B/ principles of the methods used + key results; C/ conclusions (commercial and environmental impacts)

3/ remove abbreviations, kindly understand that terminology such as "PDR18-CAMEVAR”; “FP22-CAMEPRO” and "V1, V2, V3, V4, V6, V7, V8, V11" brings nothing to our readers, please understand that the purpose of the Abstract is to explain to all readers (including those from other disciplines) what the paper is about

4/ better highlight the global impact and urgency of your findings, clearly indicate how will our international audience benefit from these revelations (better address our international audience of readers, make sure the manuscript is not limited to Spain)

Introduction:

5/ remove all clusters of references to avoid reference overkill (prefer only 1 reference to support 1 claim)

6/ deeper review the latest trends in oil-crops, refer to paper "Pressure Shockwaves to Enhance Oil Extraction from Jatropha Curcas L."

7/ go straight to the point and more in depth, write more technically (always provide corresponding numbers), significantly condensate all the text by reducing ballast phrases and cliché

8/ the economic reality should not be ignored, refer to papers "The analysis of investment into industries based on portfolio managers" and "Clusters in Transition to Circular Economy: Evaluation of Relation"

9/ the research hypothesis could be stated more clearly, condensate the research hypothesis into 1 short statement or question that will be subsequently confirmed or refuted, make sure the urgency and significance of the research hypothesis was justified in its environmental - economic nexus

Materials and Methods:

10/ the method must be presented in such a way that it can be reproduced anytime, by anyone, anywhere (do not create obstacles like referring to specific location etc.)

11/ complexity of phoshorus availability to organisms should be better explained, refer to Fig. 1 in paper "Novel sorbent shows promising financial results on P recovery from sludge water"

12/ please understand that the methodology must be described in a completely unambiguous way that does not allow for multiple interpretations (everyone who reads this chapter should get very precise instructions on how to repeat your procedure to achieve exactly the same results)

13/ each material/reactant and apparatus used needs to be presented in detail (serial number, setup, process parameters, manufacturer, country of origin, purity etc.)

14/ provide cost breakdown or at least some simplified financial analysis if you are about to argue that this concept is realistic

Results and discussion:

15/ avoid data overkill, present only the most most industrially important results

16/ show more self-criticism to your work (can all the methods and results be fully trusted? what are the weaknesses of the methods used? where do the main measurement inaccuracies arise? what are the limitations from a commercial point of view? are the lessons learned transferable to other fields?)

17/ comment on (refer to papers "Financial and biotechnological assessment of new oil extraction technology" and "Novel technique to enhance the disintegration effect of the pressure waves on oilseeds")

18/ Fig. 2: provide units to the Y axis

19/ compare your results in more depth with the existing literature, identify the main deviations and try to explain the mechanisms by which they may have been caused

20/ the financial issues could be disscussed better, refer to papers "Does the life cycle affect earnings management and bankruptcy?" and "Data-driven Machine Learning and Neural Network Algorithms in the Retailing Environment: Consumer Engagement, Experience, and Purchase Behaviors"

21/ reveal the main driving mechanisms of your results, provide deeper synthesis and reveal some more original/significant findings

Conclusions:

22/ clearly indicate whether the research hypotheses tends to be confirmed or not

Author Response

REVIEWER #3 Dear author(s), there are some inspiring insights thorough the manuscript and I tend to agree on its publication. However, there are few points that needs to be quickly addressed to improve its overall communication: First, thanks to the reviewer 3 for revising the manuscript. Second, we have included a general comment for the editor and reviewer, and, also, we have answered each comment made by the reviewer (in red). We believe that many of the comments by reviewer 3 would make the article a review or a perspectives paper. We think that several of the comments are interesting but out of the real topic of this short communication. We have considered those that improve the content of the current article but not the form. We follow the specifications of communication-type papers for MDPI journals (copied below our response in blue). This manuscript is part of a larger study and what we intend is to publish the work that is being done with these new varieties and to make their characteristics known. This communication was previously submitted as a research article and the editor, after the comments of the reviewers, recommended us to upload it again in a communication format. This format according to MDPI guidelines implies that it should be short, clear, and contain preliminary results included in larger-scale works, which is exactly what this manuscript fulfills. If the editor considers that we can change the format again, we will do it without any problem. “Manuscripts submitted to Land should neither be published previously nor be under consideration for publication in another journal. The main article types are as follows: Articles: Original research manuscripts, which should comprise at least 15 pages. The journal considers all original research manuscripts provided that the work reports scientifically sound experiments and provides a substantial amount of new information. Authors should not unnecessarily divide their work into several related manuscripts, although short Communications of preliminary, but significant, results will be considered. The quality and impact of the study will be considered during peer review. Reviews: These provide concise and precise updates on the latest progress made in a given area of research and should be at least 20 pages. Systematic reviews should follow the PRISMA guidelines. Technical Note: Technical notes are brief articles focused on a new technique, method, or procedure. These should describe important modifications or unique applications for the described method. Technical notes can also be used for describing a new software tool or computational method. The structure should include an Abstract, Keywords, Introduction, Materials and Methods, Results, Discussion, and Conclusions, with a suggested minimum word count of 3500 words. Communication: Communications are short articles that present groundbreaking preliminary results or significant findings that are part of a larger study over multiple years. They can also include cutting-edge methods or experiments, and the development of new technology or materials. The structure is similar to an article and there is a suggested minimum word count of 2000 words. Data Descriptors: Containing a description of a dataset, including methods used for collecting or producing the data, where the dataset may be found, and information about its use.” Title: 1/ summarise in a clear statement why your work is important to humanity. The title summarizes the content of the communication. Abstract: 2/ strictly follow the established schema of writing academic Abstract: A/ introduction (urgency and significance of the research hypothesis); B/ principles of the methods used + key results; C/ conclusions (commercial and environmental impacts). The abstract has been corrected according to the suggestions. 3/ remove abbreviations, kindly understand that terminology such as "PDR18-CAMEVAR”; “FP22-CAMEPRO” and "V1, V2, V3, V4, V6, V7, V8, V11" brings nothing to our readers, please understand that the purpose of the Abstract is to explain to all readers (including those from other disciplines) what the paper is about. Done. 4/ better highlight the global impact and urgency of your findings, clearly indicate how will our international audience benefit from these revelations (better address our international audience of readers, make sure the manuscript is not limited to Spain). The abstract has been corrected according to the suggestions. Introduction: 5/ remove all clusters of references to avoid reference overkill (prefer only 1 reference to support 1 claim). We believe that this citation format is widely accepted in scientific communications and articles e.g.: https://www.science.org/doi/10.1126/science.abf0869 6/ deeper review the latest trends in oil-crops, refer to paper "Pressure Shockwaves to Enhance Oil Extraction from Jatropha Curcas L." We have included citations newest than this one proposed by reviewer. We think the suggested paper by the reviewer shouldn’t be cited in our manuscript because it is related to Jatropha curcas which is inside the Euphorbiaceae family. 7/ go straight to the point and more in depth, write more technically (always provide corresponding numbers), significantly condensate all the text by reducing ballast phrases and cliché. The text has been reviewed and we do not agree with the perception of the reviewer. 8/ the economic reality should not be ignored, refer to papers "The analysis of investment into industries based on portfolio managers" and "Clusters in Transition to Circular Economy: Evaluation of Relation" Our article is not about economy, investment, or circular economy. It is about the characteristics of a promising crop such as camelina. 9/ the research hypothesis could be stated more clearly, condensate the research hypothesis into 1 short statement or question that will be subsequently confirmed or refuted, make sure the urgency and significance of the research hypothesis was justified in its environmental - economic nexus. The objective is to try in the field the newly developed varieties and to know their characteristics. We think that it is clearly shown in the introduction. Materials and Methods: 10/ the method must be presented in such a way that it can be reproduced anytime, by anyone, anywhere (do not create obstacles like referring to specific location etc.). The experiments were carried out in specific farms, so they must be highlighted in the methodology. Omitting this information is not being clear about the study area. We believe that this is not an obstacle to repeat sowing in other parts of the world. 11/ complexity of phoshorus availability to organisms should be better explained, refer to Fig. 1 in paper "Novel sorbent shows promising financial results on P recovery from sludge water". This is out of scope. We do not talk about soil typology at any time. 12/ please understand that the methodology must be described in a completely unambiguous way that does not allow for multiple interpretations (everyone who reads this chapter should get very precise instructions on how to repeat your procedure to achieve exactly the same results). We think that the methodology is clearly written. 13/ each material/reactant and apparatus used needs to be presented in detail (serial number, setup, process parameters, manufacturer, country of origin, purity etc.) We clearly cite that the experiments were carried out in other labs (we indicate the techniques used). We also explain that the varieties are developed here in Madrid (Camelina Company S.L.), etc. 14/ provide cost breakdown or at least some simplified financial analysis if you are about to argue that this concept is realistic. This article is not about finance or economics, so we think that this comment is out of the scope. Results and discussion: 15/ avoid data overkill, present only the most most industrially important results. 16/ show more self-criticism to your work (can all the methods and results be fully trusted? what are the weaknesses of the methods used? where do the main measurement inaccuracies arise? what are the limitations from a commercial point of view? are the lessons learned transferable to other fields?). Answering all this is outside the scope of the article. These questions are for a longer article reviewing other issues affecting agriculture in general and all published work on crops and land use. This is a descriptive paper (short communication) where we try to show the characteristics of nine new varieties developed in Madrid with our collaborators. 17/ comment on (refer to papers "Financial and biotechnological assessment of new oil extraction technology" and "Novel technique to enhance the disintegration effect of the pressure waves on oilseeds"). It is not a financial paper or a methodology paper. 18/ Fig. 2: provide units to the Y axis. This figure shows different ratios that refer to percentages as indicated in the Table 2. We have clarified the figure caption. 19/ compare your results in more depth with the existing literature, identify the main deviations and try to explain the mechanisms by which they may have been caused. The results are compared to other papers cited in the body text. 20/ the financial issues could be disscussed better, refer to papers "Does the life cycle affect earnings management and bankruptcy?" and "Data-driven Machine Learning and Neural Network Algorithms in the Retailing Environment: Consumer Engagement, Experience, and Purchase Behaviors". This paper is not a review of financial issues, this comment is out of scope. 21/ reveal the main driving mechanisms of your results, provide deeper synthesis and reveal some more original/significant findings. We think that developing new varieties of crops is a significant finding. Of course, more work is needed to know properly these varieties, but this paper allows us to show to the scientific community these new crops. Conclusions: 22/ clearly indicate whether the research hypotheses tends to be confirmed or not. This article is a communication which includes new data of different varieties. We think that the conclusions are clearly written.

Reviewer 4 Report (New Reviewer)

The article contained several yield and quality parameters to prove the significant importance of Camelina for cultivation in Spain. I shall suggest adding significant letters to highlight the significant differences between different cultivars. Also, I shall suggest converting some tables into bar charts like Table 4 and 5. 

Regards

Author Response

REVIEWER #4 The article contained several yield and quality parameters to prove the significant importance of Camelina for cultivation in Spain. I shall suggest adding significant letters to highlight the significant differences between different cultivars. Also, I shall suggest converting some tables into bar charts like Table 4 and 5. Regards Thanks for your comments. We prefer to keep the tables, where the readers can see properly all the data included. If the editor recommends us to change them, we will do it.

Round 2

Reviewer 2 Report (New Reviewer)

I have no other comments.  I sincerely hope the authors understand that my review was not holding them back, but trying to help improve this short manuscript, especially for better communication in the camelina research community.

This manuscript is a resubmission of an earlier submission. The following is a list of the peer review reports and author responses from that submission.

Round 1

Reviewer 1 Report

Manuscript ID: land-2005179

Type of manuscript: Article
Title:
Products and by-products characterization of Camelina sativa (L.) Crantz varieties as a promising crop in central Spain

Authors: Montero Muñoz Iris , Mostaza-Colado David , Capuano Aníbal , Mauri Ablanque Pedro V. *

Comments

The manuscript presented for revision is interesting. Similar works and studies, regarding the characteristics of seeds, have been reported in recent years. But these publications mainly deal with seed characteristics. There is a lack of data on the composition and use of straw.

But, unfortunately, the authors did not avoid mistakes. They also used literature sources containing already outdated data. Scientific progress, progress in agrotechnics and breeding works lead to the creation of new varieties of oil seeds (e.g. rapeseeds!) that are now used in field crops.

I have a few comments about the manuscript:

 Title and abstract:

The title should be clarified. What do the authors understand by the term "product" and what do "by-product"? In chapter 3.3. Straw and seed chemical composition (line 223), as "by-product" - seeds and straw are discussed. If "product" is seeds, then in my opinion, many readers may understand "by-product" as pomace, cake or meal, or by-products of seed pressing. Please explain it.

Introduction:

The introduction should also mention the use of camelina oil as an edible oil used in human diets. For example, in Poland, cold-pressed camelina oil is a traditional product used in traditional dishes. Cold-pressed camelina oil has been entered into the EU register of Traditional Specialities Guaranteed in 2009 (Regulation (EC) No. 506/2009 of June 2009). (e.g. Ratusz et al. (2016): Oxidative stability of camelina (Camelina sativa L.) oil using PDSC and Rancimat method; or Ratusz et al. (2018): Bioactive Compounds, Nutritional Quality and Oxidative Stability of Cold-Pressed Camelina (Camelina sativa L.) Oils).

Materials and methods:

What methods were used to determine the fat content and fatty acid profile of the seeds? What methods were used to evaluate the nutrient composition of seeds and straw Camelina sativa?

Results and discussion:

·       Fatty acids. The fatty acid profile of camelina oil makes it a good potential material for the production of biodiesel, but on the other hand it is a very good, valuable edible oil. Nutritional Quality Index of camelina oil are very favorable, and at the same time the oxidative stability of this oil is much higher than that of e.g. linseed oil. Please use the works of Ratusz et al 2016 and 2018 for discussion.

·       Line. 213- 215 "The oilseed more similar to C. sativa is the canola, but the erucic acid content, considered to be a limiting factor for human consumption, in the oil of C. sativa ranged from 2.68% to 3.17%, while in the rapeseed is 45% to 54% [61]".  It is not true, it is false! Rapeseed oil with such erucic acid content is technical oil!!.  Such oil cannot be used for food purposes. Data from [61] are over 50 years old!!!! This data is out of date. Rapeseed oil from "00" seeds has less than 2% erucic acid !!! Please replace this source with newer data (eg Codex Alimentarius, or more recent publications on rapeseed oil (eg Prof. Wroniak).

·       Line 223: "Regarding the by-products (seed and straw) of Camelina sativa" Please explain what is a "product" and what is a "by-product"?

Author Response

Response to Reviewer 1 Comments

Point 1: Title and abstract:

The title should be clarified. What do the authors understand by the term "product" and what do "by-product"? In chapter 3.3. Straw and seed chemical composition (line 223), as "by-product" - seeds and straw are discussed. If "product" is seeds, then in my opinion, many readers may understand "by-product" as pomace, cake or meal, or by-products of seed pressing. Please explain it.

Response 1: As suggested by the reviewer, we have decided to substitute the words products and by-products by seed and straw to avoid confusion. In our case, the seed was the product, and the straw was the by-product.

Point 2: Introduction:

The introduction should also mention the use of camelina oil as an edible oil used in human diets. For example, in Poland, cold-pressed camelina oil is a traditional product used in traditional dishes. Cold-pressed camelina oil has been entered into the EU register of Traditional Specialities Guaranteed in 2009 (Regulation (EC) No. 506/2009 of June 2009). (e.g. Ratusz et al. (2016): Oxidative stability of camelina (Camelina sativa L.) oil using PDSC and Rancimat method; or Ratusz et al. (2018): Bioactive Compounds, Nutritional Quality and Oxidative Stability of Cold-Pressed Camelina (Camelina sativa L.) Oils).

Response 2: Mention to the use of camelina oil has been incorporated in the introduction.

Point 3: Materials and methods:

What methods were used to determine the fat content and fatty acid profile of the seeds? What methods were used to evaluate the nutrient composition of seeds and straw Camelina sativa?

Response 3: We have added the methods used by the companies SETNA and SERIDA.

Point 4: Results and discussion:

  • Point 4.1. Fatty acids. The fatty acid profile of camelina oil makes it a good potential material for the production of biodiesel, but on the other hand it is a very good, valuable edible oil. Nutritional Quality Index of camelina oil are very favorable, and at the same time the oxidative stability of this oil is much higher than that of e.g. linseed oil. Please use the works of Ratusz et al 2016 and 2018 for discussion.
  • Response 4.1:
  • Point 4.2. 213- 215 "The oilseed more similar to C. sativa is the canola, but the erucic acid content, considered to be a limiting factor for human consumption, in the oil of C. sativa ranged from 2.68% to 3.17%, while in the rapeseed is 45% to 54% [61]". It is not true, it is false! Rapeseed oil with such erucic acid content is technical oil!!. Such oil cannot be used for food purposes. Data from [61] are over 50 years old!!!! This data is out of date. Rapeseed oil from "00" seeds has less than 2% erucic acid !!! Please replace this source with newer data (eg Codex Alimentarius, or more recent publications on rapeseed oil (eg Prof. Wroniak).
  • Response 4.2: Due to the fact that we have added new references, the number 61 cited by the reviewer becomes number 63. The new reference added is from 2011.
  • 3. Line 223: "Regarding the by-products (seed and straw) of Camelina sativa" Please explain what is a "product" and what is a "by-product"?
  • Response 4.3: We have solved this problem using only seed or straw words.

Reviewer 2 Report

The paper is well presented. However, the lack of any references to EXPERIMENTAL DESIGN and STATISTICAL ANALYSIS is a major shortcoming of the paper. As the plots were marked out were they replicated? How were the data analysed? were they subjected to any statistical analysis like ANOVA?

Author Response

Response to Reviewer 2 Comments

Point 1: The paper is well presented. However, the lack of any references to EXPERIMENTAL DESIGN and STATISTICAL ANALYSIS is a major shortcoming of the paper. As the plots were marked out were they replicated? How were the data analysed? were they subjected to any statistical analysis like ANOVA?

Response 1: Our results only include data of two seasons, and one of them covers the pandemic confinement period in Spain. That is why we consider that we do not have enough data to perform a full statistical analysis. Nevertheless, we believe that our preliminary results are relevant for the field, especially for the Iberian Peninsula, where camelina crop has been recently introduced. They are also the foundation for future comparisons with other varieties which are being currently introduced. Considering the reviewer’s suggestions, we have included some descriptive statistical data in Table 1 and 2, and performed some statistical analysis when possible (see lines 116-120). Finally, the methodology has been updated as well.

Point 2: At this point the authors need to describe the experimental design and the number of replications - this is important for statistical analysis.

Response 2: As we have previously explained, we do not have replications of each plot in this first set of results. Our intention is to provide a somewhat more descriptive than analytical paper covering the main topics related to camelina crops with our two first seasons data. We had one plot of 1 ha per variety, in total 9 ha. We have included a statistical analysis to see if there are differences in the yield of the two seasons included or between the varieties.

Point 3: Revise the whole document and write in past tense: Line 94 (fertilized) and Line 89 (add and sow).

Response 3: The text has been reviewed and changes have been applied following the indications of the reviewer when necessary (i.e. lines 89 and 94).

Point 4: Yield section: Was this based on statistical analysis? If so then you need to attend to my earlier comment on Experimental Design.

Response 4: This subject has been already addressed in the previous answers to the reviewer.

Round 2

Reviewer 2 Report

Comments noted.